# Preference Transformer: Modeling Human Preferences using Transformers for RL

Changyeon Kim[1*] Jongjin Park[1*] Jinwoo Shin[1] Honglak Lee[2,3] Pieter Abbeel[4] Kimin Lee[5]

[1]KAIST [2]University of Michigan [3]LG AI Research [4]UC Berkeley [5]Google Research

## Abstract

Preference-based reinforcement learning (RL) provides a framework to train agents using human preferences between two behaviors. However, preference-based RL has been challenging to scale since it requires a large amount of human feedback to learn a reward function aligned with human intent. In this paper, we present Preference Transformer, a neural architecture that models human preferences using transformers. Unlike prior approaches assuming human judgment is based on the Markovian rewards which contribute to the decision equally, we introduce a new preference model based on the weighted sum of non-Markovian rewards. We then design the proposed preference model using a transformer architecture that stacks causal and bidirectional self-attention layers. We demonstrate that Preference Transformer can solve a variety of control tasks using real human preferences, while prior approaches fail to work. We also show that Preference Transformer can induce a well-specified reward and attend to critical events in the trajectory by automatically capturing the temporal dependencies in human decision-making. Code is available on the project website: https://sites.google.com/view/preference-transformer.

## 1 Introduction

Reinforcement learning (RL) has been successful in solving sequential decision-making problems in various domains where a suitable reward function is available (Mnih et al., 2015; Silver et al., 2017; Berner et al., 2019; Vinyals et al., 2019). However, reward engineering poses a number of challenges. It often requires extensive instrumentation (*e.g.*, thermal cameras (Schenck & Fox, 2017), accelerometers (Yahya et al., 2017), or motion trackers (Peng et al., 2020)) to design a dense and precise reward. Also, it is hard to evaluate the quality of outcomes in a single scalar since many problems have multiple objectives. For example, we need to care about many objectives like velocity, energy spent, and torso verticality to achieve stable locomotion (Tassa et al., 2012; Faust et al., 2019). It requires substantial human effort and extensive task knowledge to aggregate multiple objectives into a single scalar.

To avoid reward engineering, there are various ways to learn the reward function from human data, such as real-valued feedback (Knox & Stone, 2009; Daniel et al., 2014), expert demonstrations (Ng et al., 2000; Abbeel & Ng, 2004), preferences (Akrour et al., 2011; Wilson et al., 2012; Sadigh et al., 2017) and language instructions (Fu et al., 2019; Nair et al., 2022). Especially, research interest in preference-based RL (Akrour et al., 2012; Christiano et al., 2017; Lee et al., 2021b) has increased recently since making relative judgments (*e.g.*, pairwise comparison) is easy to provide yet information-rich. By learning the reward function from human preferences between trajectories, recent work has shown that the agent can learn novel behaviors (Christiano et al., 2017; Stiennon et al., 2020) or avoid reward exploitation (Lee et al., 2021b).

However, existing approaches still require a large amount of human feedback, making it hard to scale up preference-based RL to various applications. We hypothesize this difficulty originated from common underlying assumptions in preference modeling used in most prior work. Specifically, prior work commonly assumes that (a) the reward function is Markovian (*i.e.*, depending only on the current state and action), and (b) human evaluates the quality of a trajectory (agent's behavior) based on the sum of rewards with equal weights. These assumptions can be flawed due to the

---

*Equal contribution. Correspondence to: {changyeon.kim, jongjin.park}@kaist.ac.kr

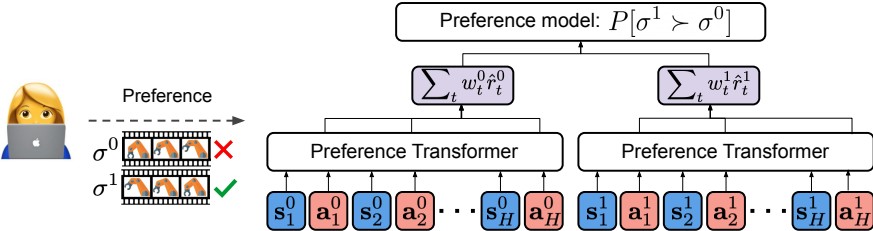

Figure 1: Illustration of our framework. Given a preference between two trajectory segments $(\sigma^0, \sigma^1)$, Preference Transformer generates non-Markovian rewards $\hat{r}_t$ and their importance weights $w_t$ over each segment. We then model the preference predictor based on the weighted sum of non-Markovian rewards (*i.e.*, $\sum_t w_t \hat{r}_t$), and align it with human preference.

following reasons. First, there are various tasks where rewards depend on the visited states (*i.e.*, non-Markovian) since it is hard to encode all task-relevant information into the state (Bacchus et al., 1996; 1997). This can be especially true in preference-based learning since the trajectory segment is provided to the human sequentially (*e.g.*, a video clip (Christiano et al., 2017; Lee et al., 2021b)), enabling earlier events to influence the ratings of later ones. In addition, since humans are highly sensitive to remarkable moments (Kahneman, 2000), credit assignment within the trajectory is required. For example, in the study of human attention on video games using eye trackers (Zhang et al., 2020), the human player requires a longer reaction time and multiple eye movements on important states that can lead to a large reward or penalty, in order to make a decision.

In this paper, we aim to propose an alternative preference model that can overcome the limitations of common assumptions in prior work. To this end, we introduce the new preference model based on the weighted sum of non-Markovian rewards, which can capture the temporal dependencies in human decisions and infer critical events in the trajectory. Inspired by the recent success of transformers (Vaswani et al., 2017) in modeling sequential data (Brown et al., 2020; Chen et al., 2021), we present Preference Transformer, a transformer-based architecture for designing the proposed preference model (see Figure 1). Preference Transformer takes a trajectory segment as input, which allows extracting task-relevant historical information. By stacking bidirectional and causal self-attention layers, Preference Transformer generates non-Markovian rewards and importance weights as outputs. We then utilize them to define the preference model.

We highlight the main contributions of this paper below:

- We propose a more generalized framework for modeling human preferences based on a weighted sum of non-Markovian rewards.
- We present Preference Transformer, a transformer-based architecture that consists of our novel preference attention layer designed for the proposed framework.
- Preference Transformer enable RL agents to solve complex navigation, locomotion tasks from D4RL (Fu et al., 2020) benchmarks and robotic manipulation tasks from Robomimic (Mandlekar et al., 2021) benchmarks by learning a reward function from *real human* preferences.
- We analyze the learned reward function and importance weights, showing that Preference Transformer can induce a well-specified reward and capture critical events within a trajectory.

## 2 RELATED WORK

**Preference-based reinforcement learning**. Recently, various methods have utilized human preferences to train RL agents without reward engineering (Akrour et al., 2012; Christiano et al., 2017; Ibarz et al., 2018; Stiennon et al., 2020; Lee et al., 2021b;c; Nakano et al., 2021; Wu et al., 2021; III & Sadigh, 2022; Knox et al., 2022; Ouyang et al., 2022; Park et al., 2022; Verma & Metcalf, 2022). Christiano et al. (2017) showed that preference-based RL can effectively solve complex control tasks using deep neural networks. To improve the feedback-efficiency, several methods, such as pre-training (Ibarz et al., 2018; Lee et al., 2021b), data augmentation (Park et al., 2022), exploration (Liang et al., 2022), and meta-learning (III & Sadigh, 2022), have been proposed. Preference-based RL also has been successful in fine-tuning large-scale language models (such as GPT-3; Brown et al. 2020) for hard tasks (Stiennon et al., 2020; Wu et al., 2021; Nakano et al., 2021; Ouyang et al.,

2022). Transformer-based models (*i.e.*, pre-trained language models) are used as a reward function in these approaches due to partial observability of language inputs, but we utilize transformers with a new preference modeling for control tasks.

**Transformer for reinforcement learning and imitation learning**. Transformers (Vaswani et al., 2017) have been studied for various purposes in RL (Vinyals et al., 2019; Zambaldi et al., 2019; Parisotto et al., 2020; Chen et al., 2021; Janner et al., 2021). It has been observed that sample-efficiency and generalization ability can be improved by modeling RL agents using transformers in complex environments, such as StarCraft (Vinyals et al., 2019; Zambaldi et al., 2019) and DMLab-30 (Parisotto et al., 2020) benchmarks. For offline RL, Chen et al. (2021) and Janner et al. (2021) also leveraged the transformers by formulating RL problems in the context of the sequential modeling problem. Additionally, transformers have been applied successfully in imitation learning (Dasari & Gupta, 2020; Mandi et al., 2022; Reed et al., 2022). Dasari & Gupta (2020) and Mandi et al. (2022) demonstrated the generalization ability of transformers in one-shot imitation learning (Duan et al., 2017), and Reed et al. (2022) utilized the transformers for multi-domain and multi-task imitation learning. In this work, we demonstrate that transformers also can be useful in improving the efficiency of preference-based learning.

**Non-Markovian reward learning**. Non-Markovian reward (Bacchus et al., 1996; 1997) has been studied for dealing with the realistic reward setting in which reward depends on the visited states, or reward is delayed or even given at the end of each episode. Several recent work focuses on return decomposition where an additional model is trained under the objective of predicting trajectory return with a given state-action sequence and used for reward redistribution. Arjona-Medina et al. (2019) and Early et al. (2022) used LSTM (Hochreiter & Schmidhuber, 1997) for capturing sequential information in reward learning. Gangwani et al. (2020) and Ren et al. (2022) proposed simplified approaches without any prior task knowledge, under the assumption that episodic return is distributed uniformly in the timestep. In this work, we adopt a transformer-based reward model for learning non-Markovian rewards, which can capture the temporal dependencies in human decisions.

## 3 PRELIMINARIES

We consider the reinforcement learning (RL) framework where an agent interacts with an environment in discrete time (Sutton & Barto, 2018). Formally, at each timestep $t$, the agent receives the current state $\mathbf{s}_t$ from the environment and chooses an action $\mathbf{a}_t$ based on its policy $\pi$. The environment gives a reward $r(\mathbf{s}_t, \mathbf{a}_t)$, and the agent transitions to the next state $\mathbf{s}_{t+1}$. The goal of RL is to learn a policy that maximizes the expected return, $\mathcal{R}_t = \sum_{k=0}^{\infty} \gamma^k r(\mathbf{s}_{t+k}, \mathbf{a}_{t+k})$, which is defined as a discounted cumulative sum of the reward with discount factor $\gamma$.

In many applications, it is difficult to design a suitable reward function capturing human intent. Preference-based RL (Akrour et al., 2011; Pilarski et al., 2011; Wilson et al., 2012; Christiano et al., 2017) addresses this issue by learning a reward function from human preferences. Similar to Wilson et al. (2012) and Christiano et al. (2017), we consider preferences over two trajectory segments of length $H$, $\sigma = \{(\mathbf{s}_1, \mathbf{a}_1), ..., (\mathbf{s}_H, \mathbf{a}_H)\}$. Given a pair of segments $(\sigma^0, \sigma^1)$, a (human) teacher indicates which segment is preferred, i.e., $y \in \{0, 1, 0.5\}$. The preference label $y = 1$ indicates $\sigma^1 \succ \sigma^0$, $y = 0$ indicates $\sigma^0 \succ \sigma^1$, and $y = 0.5$ implies an equally preferable case, where $\sigma^i \succ \sigma^j$ denotes the event that the segment $i$ is preferable to the segment $j$. Each feedback is stored in a dataset of preferences $\mathcal{D}$ as a triple $(\sigma^0, \sigma^1, y)$.

To obtain a reward function $\hat{r}$ parameterized by $\psi$, most prior work (Christiano et al., 2017; Ibarz et al., 2018; Lee et al., 2021b;c; III & Sadigh, 2022; Park et al., 2022) defines a preference predictor following the Bradley-Terry model (Bradley & Terry, 1952):

$$P[\sigma^1 \succ \sigma^0; \psi] = \frac{\exp\left(\sum_t \hat{r}(\mathbf{s}_t^1, \mathbf{a}_t^1; \psi)\right)}{\exp\left(\sum_t \hat{r}(\mathbf{s}_t^1, \mathbf{a}_t^1; \psi)\right) + \exp\left(\sum_t \hat{r}(\mathbf{s}_t^0, \mathbf{a}_t^0; \psi)\right)}. \tag{1}$$

Then, given a dataset of preferences $\mathcal{D}$, the reward function $\hat{r}$ is updated by minimizing the cross-entropy loss between this preference predictor and the actual human labels:

$$\mathcal{L}^{\text{CE}}(\psi) = - \mathop{\mathbb{E}}_{(\sigma^0, \sigma^1, y) \sim \mathcal{D}} \left[ (1 - y) \log P[\sigma^0 \succ \sigma^1; \psi] + y \log P[\sigma^1 \succ \sigma^0; \psi] \right]. \tag{2}$$

One can update a policy $\pi$ using any RL algorithm such that it maximizes the expected returns with respect to the learned reward.

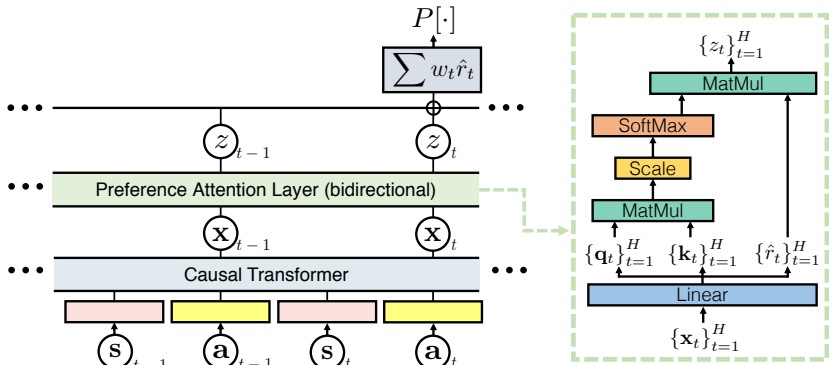

Figure 2: Overview of Preference Transformer. We first construct hidden embeddings $\{\mathbf{x}_t\}$ through the causal transformer, where each represents the context information from the initial timestep to timestep $t$. The preference attention layer with a bidirectional self-attention computes the non-Markovian rewards $\{\hat{r}_t\}$ and their convex combinations $\{z_t\}$ from those hidden embeddings, then we aggregate $\{z_t\}$ for modeling the weighted sum of non-Markovian rewards $\sum_t w_t \hat{r}_t$.

## 4    PREFERENCE TRANSFORMER

In this section, we present Preference Transformer (PT), the transformer architecture for modeling human preferences (see Figure 2 for the overview). First, we introduce a new preference predictor $P[\sigma^1 \succ \sigma^0]$ based on a weighted sum of non-Markovian rewards in Section 4.1, which can reflect the long-term context of the agent's behaviors and capture the critical events in the trajectory segment. We then describe a novel transformer-based architecture to model the proposed preference predictor in Section 4.2.

### 4.1    PREFERENCE MODELING

As mentioned in Section 3, most prior work assumes that the reward is Markovian (*i.e.*, depending only on current state and action), and human evaluates the quality of trajectory segment based on the sum of rewards with equal weight. Based on these assumptions, a preference predictor $P[\sigma^1 \succ \sigma^0]$ is defined as equation 1. However, this formulation has several limitations in modeling real human preferences. First, in many cases, it is hard to specify the tasks using the Markovian reward (Bacchus et al., 1996; 1997; Early et al., 2022). Furthermore, credit assignment within the trajectory can be required since human is sensitive to remarkable moments (Kahneman, 2000).

To address the above issues, we introduce a new preference predictor that assumes the probability of preferring a segment depends exponentially on the weighted sum of non-Markovian rewards:

$$P[\sigma^1 \succ \sigma^0; \psi] = \frac{\exp\left(\sum_t w\left(\{(\mathbf{s}_i^1, \mathbf{a}_i^1)\}_{i=1}^{H}; \psi\right)_t \cdot \hat{r}\left(\{(\mathbf{s}_i^1, \mathbf{a}_i^1)\}_{i=1}^{t}; \psi\right)\right)}{\sum_{j \in \{0,1\}} \exp\left(\sum_t w\left(\{(\mathbf{s}_i^j, \mathbf{a}_i^j)\}_{i=1}^{H}; \psi\right)_t \cdot \hat{r}\left(\{(\mathbf{s}_i^j, \mathbf{a}_i^j)\}_{i=1}^{t}; \psi\right)\right)}. \quad (3)$$

To capture the temporal dependencies, we consider a non-Markovian reward function $\hat{r}$, which receives the full preceding sub-trajectory $\{(\mathbf{s}_i, \mathbf{a}_i)\}_{i=1}^{t}$ as inputs. The importance weight $w$ is also introduced as a function of the entire trajectory segment $\{(\mathbf{s}_i, \mathbf{a}_i)\}_{i=1}^{H}$, which makes our preference predictor able to perform the credit assignment within the segment. We remark that our formulation is a generalized version of the conventional design; if the reward function only depends on the current state-action pair and the importance weight is always 1, our preference predictor would be equivalent to the standard model in equation 1.

### 4.2    ARCHITECTURE

In order to model the preference predictor as in equation 3, we propose a new transformer architecture, called Preference Transformer, that consists of the following learned components:

**Causal transformer**. We use the transformer network (Vaswani et al., 2017) as the backbone inspired by its superiority in modeling sequential data (Brown et al., 2020; Ramesh et al., 2021;

Janner et al., 2021; Chen et al., 2021). Specifically, we use the GPT architecture (Radford et al., 2018), *i.e.*, the transformer architecture with causally masked self-attention. Given trajectory segment $\sigma = \{(\mathbf{s}_1, \mathbf{a}_1), \cdots, (\mathbf{s}_H, \mathbf{a}_H)\}$ of length $H$, we generate $2H$ input embeddings (for state and action), which are learned by a linear layer followed by layer normalization (Ba et al., 2016). A shared positional embedding (*i.e.*, state and action at the same timestep share the same positional embedding) is learned and added to each input embedding. The input embeddings are then fed into the causal transformer network, and it produces output embeddings $\{\mathbf{x}_t\}_{t=1}^H$ such that $t$-th output depends on input embeddings up to $t$.

**Preference attention layer**. To model the preference predictor using the weighted sum of the non-Markovian rewards as defined in equation 3, we introduce a preference attention layer. As shown in Figure 2, the preference attention layer receives the hidden embeddings from the causal transformer $\{\mathbf{x}_t\}_{t=1}^H$ and generates rewards $\hat{r}$ and importance weights $w_t$. Formally, $t$-th input $\mathbf{x}_t$ is mapped via linear transformations to a key $\mathbf{k}_t \in \mathbb{R}^d$, query $\mathbf{q}_t \in \mathbb{R}^d$, and value $\hat{r}_t \in \mathbb{R}$, where $d$ is the embedding dimension. We remark that $t$-th value in self-attention, *i.e.*, $\hat{r}_t$, is considered as modeling a non-Markovian reward $\hat{r}(\{(\mathbf{s}_i, \mathbf{a}_i)\}_{i=1}^t)$ since hidden embedding $\mathbf{x}_t$ only depends on previous inputs in a trajectory segment. Specifically, at each timestep $t$, we use $t$ state-action pairs $\{(\mathbf{s}_i, \mathbf{a}_i)\}_{i=1}^t$ for approximating the reward $\hat{r}_t$ during the training.

Following the self-attention (Vaswani et al., 2017), the $i$-th output $z_i$ is defined as a convex combination of the values with attention weights from the $i$-th query and keys:

$$z_i = \sum_{t=1}^H \texttt{softmax}(\{\langle \mathbf{q}_i, \mathbf{k}_{t'} \rangle\}_{t'=1}^H)_t \cdot \hat{r}_t.$$

Then, the weighted sum of rewards can be computed by the average of outputs $\{z_i\}_{i=1}^H$ as follows:

$$\frac{1}{H}\sum_{i=1}^H z_i = \frac{1}{H}\sum_{i=1}^H \sum_{t=1}^H \texttt{softmax}(\{\langle \mathbf{q}_i, \mathbf{k}_{t'} \rangle\}_{t'=1}^H)_t \cdot \hat{r}_t = \sum_{t=1}^H w_t \hat{r}_t \qquad (4)$$

where $w_t = \frac{1}{H}\sum_{i=1}^H \texttt{softmax}(\{\langle \mathbf{q}_i, \mathbf{k}_{t'} \rangle\}_{t'=1}^H)_t$. Here, $w_t$ corresponds to modeling importance weight $w(\{(\mathbf{s}_i, \mathbf{a}_i)\}_{i=1}^H)_t$ in equation 3 because this preference attention is not causally masked and thus depends on the full sequence (*i.e.*, bidirectional self-attention). In summary, we model the weighted sum of non-Markovian rewards by taking the average of outputs from transformer networks and assume that the probability of preferring the segment is proportional to it. The complete architecture of Preference Transformer is shown in Figure 2.

### 4.3 TRAINING AND INFERENCE

**Training**. We train Preference Transformer by minimizing the cross-entropy loss in equation 2 given a dataset of preferences $\mathcal{D}$. By aligning a preference predictor modeled by Preference Transformer with human labels, we find that Preference Transformer can induce a suitable reward function and capture important events in the trajectory segments (see Figure 3 for supporting results).

**Inference**. For RL training, all state-action pairs are labeled with the learned reward function. Because we train the non-Markovian reward function, we provide $H$ past transitions $(\mathbf{s}_{t-H+1}, \mathbf{a}_{t-H+1}, \cdots, \mathbf{s}_t, \mathbf{a}_t)$ to Preference Transformer and use $t$-th value $\hat{r}_t$ from the preference attention layer as a reward at timestep $t$.

## 5 EXPERIMENTS

We design our experiments to investigate the following:

- Can Preference Transformer solve complex control tasks using real human preferences?
- Can Preference Transformer induce a well-aligned reward and attend to critical events?
- How well does Preference Transformer perform with synthetic preferences (*i.e.*, scripted teacher settings)?

Table 1: Averaged normalized scores of IQL on AntMaze, Gym-Mujoco locomotion tasks, and success rate on Robosuite manipulation tasks with different reward functions. Using the same dataset of preferences from real human teachers, we train Preference Transformer (PT), MLP-based Markovian reward (MR; Christiano et al. 2017; Lee et al. 2021b), and LSTM-based non-Markovian reward (NMR; Early et al. 2022). The result shows the average and standard deviation averaged over 8 runs.

| Dataset | IQL with task reward | IQL with preference learning | | |
|---|---|---|---|---|
| | | MR | NMR | PT (ours) |
| antmaze-medium-play-v2 | 73.88 $\pm$ 4.49 | 31.13 $\pm$ 16.96 | 62.88 $\pm$ 5.99 | 70.13 $\pm$ 3.76 |
| antmaze-medium-diverse-v2 | 68.13 $\pm$ 10.15 | 19.38 $\pm$ 9.24 | 20.13 $\pm$ 17.12 | 65.25 $\pm$ 3.59 |
| antmaze-large-play-v2 | 48.75 $\pm$ 4.35 | 24.25 $\pm$ 14.03 | 14.13 $\pm$ 3.60 | 42.38 $\pm$ 9.98 |
| antmaze-large-diverse-v2 | 44.38 $\pm$ 4.47 | 5.88 $\pm$ 6.94 | 0.00 $\pm$ 0.00 | 19.63 $\pm$ 3.70 |
| antmaze-v2 total | 58.79 | 20.16 | 24.29 | 49.35 |
| hopper-medium-replay-v2 | 83.06 $\pm$ 15.80 | 11.56 $\pm$ 30.27 | 57.88 $\pm$ 40.63 | 84.54 $\pm$ 4.07 |
| hopper-medium-expert-v2 | 73.55 $\pm$ 41.47 | 57.75 $\pm$ 23.70 | 38.63 $\pm$ 35.58 | 68.96 $\pm$ 33.86 |
| walker2d-medium-replay-v2 | 73.11 $\pm$ 8.07 | 72.07 $\pm$ 1.96 | 77.00 $\pm$ 3.03 | 71.27 $\pm$ 10.30 |
| walker2d-medium-expert-v2 | 107.75 $\pm$ 2.02 | 108.32 $\pm$ 3.87 | 110.39 $\pm$ 0.93 | 110.13 $\pm$ 0.21 |
| locomotion-v2 total | 84.37 | 62.43 | 70.98 | 83.72 |
| lift-ph | 96.75 $\pm$ 1.83 | 84.75 $\pm$ 6.23 | 91.50 $\pm$ 5.42 | 91.75 $\pm$ 5.90 |
| lift-mh | 86.75 $\pm$ 2.82 | 91.00 $\pm$ 4.00 | 90.75 $\pm$ 5.75 | 86.75 $\pm$ 5.95 |
| can-ph | 74.50 $\pm$ 6.82 | 68.00 $\pm$ 9.13 | 62.00 $\pm$ 10.90 | 69.67 $\pm$ 5.89 |
| can-mh | 56.25 $\pm$ 8.78 | 47.50 $\pm$ 3.51 | 30.50 $\pm$ 8.73 | 50.50 $\pm$ 6.48 |
| robosuite total | 78.56 | 72.81 | 68.69 | 74.66 |

## 5.1 SETUPS

Similar to Shin & Brown (2021), we evaluate Preference Transformer (PT) on several complex control tasks in the offline setting using D4RL (Fu et al., 2020) benchmarks and Robomimic (Mandlekar et al., 2021) benchmarks.[1] Specifically, we consider three different domains (AntMaze, Gym-Mujoco locomotion (Todorov et al., 2012; Brockman et al., 2016) from D4RL benchmarks and Robosuite robotic manipulation (Zhu et al., 2020) from Robomimic benchmarks) with different data collection schemes. For reward learning, we select queries (pairs of trajectory segments) uniformly at random from offline datasets and collect preferences from real human trainers (the authors).[2] Then, using the collected datasets of human preferences, we learn a reward function and train RL agents using Implicit Q-Learning (IQL; Kostrikov et al. 2022), a recent offline RL algorithm which achieves strong performances on D4RL benchmarks. We consider Markovian policy and value functions by following the original implementation (e.g., architecture, hyperparameters) of IQL. We additionally provide the result of offline RL experiments with the non-MDP models in Appendix G.

For evaluation, we measure expert-normalized scores respecting underlying task reward from the original benchmark (D4RL) [3] and success rate (Robomimic). As baselines, we consider the standard preference modeling based on Markovian reward (MR) or non-Markovian reward (NMR). For the MR-based model, we define the preference predictor as $P[\sigma^1 \succ \sigma^0; \psi_{MR}] = \frac{\exp\left(\sum_t \hat{r}(\mathbf{s}_t^1, \mathbf{a}_t^1; \psi_{MR})\right)}{\sum_j \exp\left(\sum_t \hat{r}(\mathbf{s}_t^j, \mathbf{a}_t^j; \psi_{MR})\right)}$ using reward function modeled with MLP similar to Christiano et al. (2017); Lee et al. (2021b). For the NMR-based model, we use the LSTM-based architecture (Early et al., 2022) as a reward function

---

[1] To focus on evaluating the performance of reward learning, we consider the offline setting. Note that similar setting assuming expert demonstrations are available is also considered in prior work (Ibarz et al., 2018).

[2] We collected 100 (medium) / 1000 (large) queries in AntMaze, 500 (medium-replay) / 100 (medium-expert) queries in Gym-Mujoco locomotion, and 100 (PH) / 500 (MH) queries in Robosuite robotic manipulations. A maximum of 10 minutes of human time was required for feedback collection in all cases except Gym-Mujoco locomotion-medium-expert and antmaze-large, which required between 1 and 2 hours of human time.

[3] Since preferences are collected by experts who are familiar with robotics and RL domains, we measure the performance with respect to task reward similar to Christiano et al. (2017).

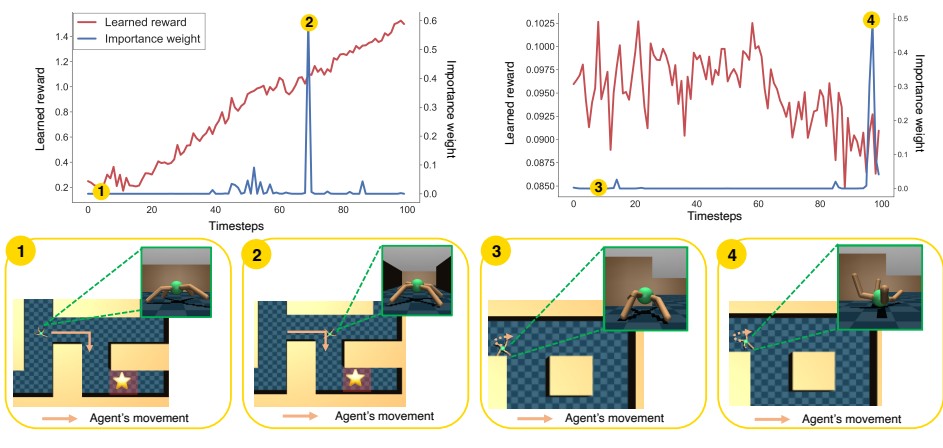

(a) Successful trajectory      (b) Failure trajectory

Figure 3: Time series of learned reward function (red curve) and importance weight (blue curve) on (a) successful trajectory segment and (b) failure trajectory segment from `antmaze-large-play-v2`. For both cases, spikes in the importance weight correspond to critical events: turning right to reach the goal (point 2), or flipping (point 4). The learned reward is also well-aligned with human intent: reward increases as the agent gets close to the goal, while it decreases when agent is flipped.

and define the preference predictor as $P[\sigma^1 \succ \sigma^0; \psi_{\text{NMR}}] = \frac{\exp\left(\sum_t \hat{r}\left(\{(\mathbf{s}_i^1, \mathbf{a}_i^1)\}_{i=1}^t; \psi_{\text{NMR}}\right)\right)}{\sum_j \exp\left(\sum_t \hat{r}\left(\{(\mathbf{s}_i^j, \mathbf{a}_i^j)\}_{i=1}^t; \psi_{\text{NMR}}\right)\right)}$.[4] Note that both baselines use the same preference modeling based on the sum of rewards with equal weight, while our method is based on the weighted sum of non-Markovian rewards modeled by transformers. We also report the performance of IQL trained with task reward from benchmarks as a reference. For all experiments, we report the mean and standard deviation across 8 runs. More experimental details (*e.g.*, task descriptions, feedback collection, and reward learning) are in Appendix B and C.

## 5.2 BENCHMARK TASKS WITH REAL HUMAN TEACHERS

Table 1 shows the performances of IQL with different reward functions. Preference Transformer consistently outperforms all baselines in almost all tasks. Especially, only our method almost matches the performance of IQL with the task reward, while baselines fail to work in hard tasks. This implies that PT can induce a suitable reward function from real human preferences and teach meaningful behaviors. In particular, there is a big gap between PT and baselines in complex tasks (*i.e.*, AntMaze) since capturing the long-term context of the agent's behaviors (*e.g.*, direction of the agent) and critical events (*e.g.*, goal location) is important in this task. These results show that our transformer-based preference model is very effective in reward learning.

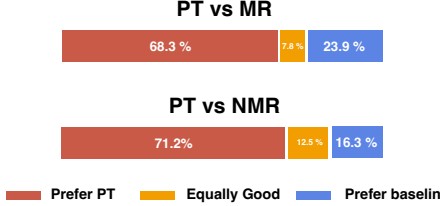

Figure 4: Averaged human evaluation results on 4 AntMaze tasks. Numbers denote the statistics of the evaluators' responses over 40 trials. PT received higher ratings compared to both MR and NMR.

**Human evaluation**. To check whether learned rewards are indeed aligned with human preferences, we also conduct a human evaluation. We generate a query (i.e., two trajectories) from agents trained with two different rewards and a human evaluator (authors) decides which trajectory is better.[5] Figure 4 shows the results comparing Preference Transformer and baselines averaged over 40 sets of

---

[4]Original work (Early et al., 2022) only considered the return prediction but we utilize the proposed LSTM-architecture in preference-based learning.

[5]Trajectories are anonymized for fair comparisons. Also, the evaluators skip the query if both agents equally fail to solve the task for meaningful comparisons.

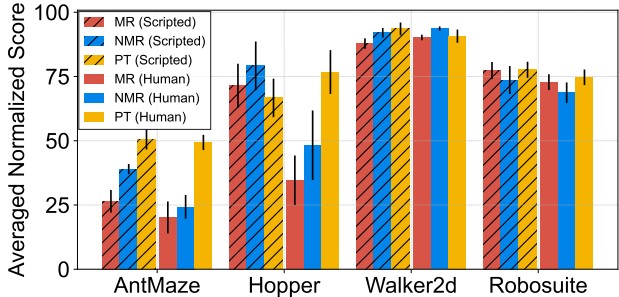

Figure 5: Averaged normalized scores of IQL with various reward functions trained from human and synthetic preferences on AntMaze, Gym-Mujoco locomotion (Hopper and Walker2d), and Robosuite robotic manipulation tasks. The result shows the mean and standard deviation averaged over 8 runs. Our method (PT) achieves strong performances on both scripted and human teachers, while the performances of baselines (MR and NMR) are significantly reduced on human teachers.

queries.[6] We observe that human evaluators prefer agents from PT compared to agents from MR or NMR, showing PT is more aligned with human preferences.

### 5.3 REWARD AND WEIGHT ANALYSIS

We evaluate whether Preference Transformer can induce a well-specified reward and capture the critical events from human preferences. Figure 3 shows the learned reward function (red curve) and importance weight (blue curve) on successful and failure trajectory segments from `antmaze-large-play-v2`. First, we find that the reward function is well-aligned with human intent. In the successful trajectory (Figure 3a), reward value increases as the agent gets close to the goal, while the failure trajectory shows low rewards since the agent struggles to escape the corner, then it flips as shown in Figure 3b. This shows that Preference Transformer can capture the context of the agent's behaviors in the reward function. We also find that importance weight shows a different trend compared to reward function. Interestingly, for both successful and failure trajectories, spikes in importance weights correspond to critical events such as turning right to reach the goal or flipping. More video examples are available in the supplementary material.

### 5.4 BENCHMARK TASKS WITH SCRIPTED TEACHERS

Similar to prior work (Christiano et al., 2017; Lee et al., 2021b;c), we also evaluate Preference Transformer using synthetic preferences from scripted teachers. We consider a deterministic teacher, which generates preferences based on task reward $r$ from the benchmark as follows: $y = i$, where $i = \arg\max_i \sum_{t=1}^{H} r(\mathbf{s}_t^i, \mathbf{a}_t^i)$. We remark that this scripted teacher is a special case of the preference model introduced in equation 1.[7]

Figure 5 shows the performances of IQL with different reward functions from both human and scripted teachers (see Appendix D for full results). Preference Transformer achieves strong performances on scripted teachers even though synthetic preferences are based on Markovian rewards which contribute to the decision equally. We expect this is because Preference Transformer can induce a better-shaped reward by utilizing historical information. Also, non-Markovian formulation (in equation 3) can be interpreted as a generalized version of Markovian formulation (in equation 1) since it can learn Markovian rewards by ignoring the history in the inputs. We also remark that baselines achieve better performances with scripted teachers compared to real human teachers on some easy tasks (*e.g.*, locomotion tasks). This implies that scripted teacher does not model real human behavior exactly, and evaluation on scripted teachers can generate misleading information. To investigate this in detail, we measure the agreement rates between human and scripted teachers. Figure 6b shows that disagreement rates are quite high since the scripted teacher can not catch

---

[6]Each set consists of 5 rollouts from 8 models trained with different random seeds.

[7]If we introduce a rationality constant (or temperature) $\beta$ to the exponential term in equation 1 and set it as $\beta \to \infty$, the preference model becomes deterministic.

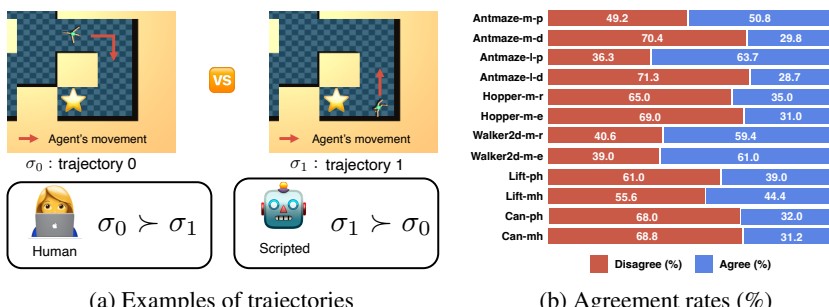

| (a) Examples of trajectories | (b) Agreement rates (%) |

Figure 6: Difference between the human and scripted teacher. (a) Examples of trajectories shown to the human and scripted teacher on AntMaze task. The human teacher provides the correct label by catching the context of behavior (*i.e.* direction) while the scripted teacher does not. (b) Agreement between human teachers and scripted teachers. We find that disagreement rates are quite high across all tasks, implying that evaluation on scripted teacher can generate misleading information.

the context of the agent's behavior correctly (see Figure 6a). Note that the disagreement between human and scripted teachers also has been observed in simple grid world domains (Knox et al., 2022). These results imply that existing preference-based RL benchmarks (Lee et al., 2021c) based on synthetic preferences may not be enough, calling for a new benchmark specially designed for preference-based RL.

## 5.5 LEARNING COMPLEX NOVEL BEHAVIORS

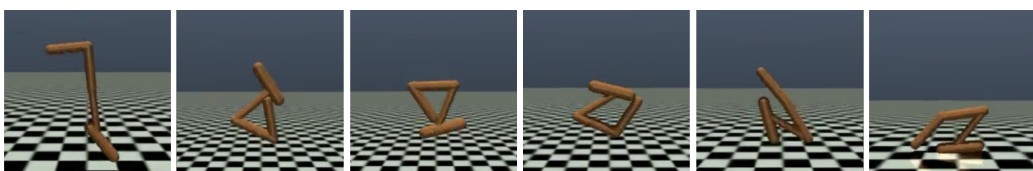

Figure 7: Six frames from a double backflip of Hopper. The agent is trained to perform a sequence of backflips using 300 queries of human feedback.

We present the effectiveness of Preference Transformer for enabling agents to learn complex and novel behaviors where a suitable reward function is difficult to design. Specifically, we demonstrate Hopper in Gym-Mujoco locomotion performing multiple backflips at each jump. This task is more challenging compared to a single backflip considered in previous preference-based RL approaches (Christiano et al., 2017; Lee et al., 2021b), as the reward function must capture non-Markovian contexts including the number of rotations. We observe that the Hopper agent learns to perform double backflip as shown in Figure 7, while the agent with Markovian reward function fails to learn it. The implementation details of Hopper backflip are provided in Appendix B, and videos of all behaviors (including *triple* backflip) are available on the project website.

## 6 DISCUSSION

In this paper, we present a new framework for modeling human preferences based on the weighted sum of non-Markovian rewards, and design the proposed framework using a transformer-based architecture. We propose a novel preference attention layer for learning the weighted sum of the non-Markovian reward function, which enables inferring the importance weight of each reward via the self-attention mechanism. Our experiments demonstrate that Preference Transformer significantly outperforms the current preference modeling methods on complex navigation, locomotion, and robotic manipulation tasks from offline RL benchmarks. In addition, we observe that the learned preference attention layer can indeed capture the events critical to the human decision. We believe that Preference Transformer is essential to scale preference-based RL (and other human-in-the-loop learning) to various applications.

## FUTURE DIRECTIONS

There are several future directions in Preference Transformer. One is to utilize the importance weights in reinforcement learning or preference-based reward learning. For example, importance weights can be utilized for sampling more informative queries, which can improve the feedback-efficiency of preference-based reward learning (Sadigh et al., 2017; Lee et al., 2021c). It is also an interesting direction to use the importance weights for stabilizing Q-learning via weighted updates (Kumar et al., 2020; Lee et al., 2021a). Combination with other preference models is another important direction for future research. For example, Knox et al. (2022) proposed a new preference model based on each segment's regret in simple grid world environments. Even though their proposed method is based on several assumptions (*e.g.*, generating successor features (Dayan, 1993; Barreto et al., 2017)), a combination with the regret-based model would be interesting.

## ETHICS STATEMENT

Unlike other domains (*e.g.*, language), control tasks require more high-quality human feedback from domain experts. Even though the quality of the dataset can be improved by training labelers, such training requires substantial effort. In addition, feedback from crowd-sourcing platforms, such as Amazon Mechanical Turk, can be noisy. We addressed these concerns by collecting human feedback from domain experts (authors) who are very familiar with robotics, RL, and target tasks. We expect that our datasets are labeled with high-quality and clean labels, and our collection strategy is closer to practice. However, at the same time, we think that evaluations using public crowd-sourcing platforms would be interesting and leave it to future work.

## REPRODUCIBILITY STATEMENT

We describe the implementation details of Preference Transformer in Appendix B, and our source code is available on the project website in the abstract. We will also publicly release the collected offline dataset with real human preferences for benchmarks.

## ACKNOWLEDGMENTS AND DISCLOSURE OF FUNDING

We would like to thank Sihyun Yu, Subin Kim, Younggyo Seo, and anonymous reviewers for providing helpful feedback and suggestions for improving our paper. This research is supported by Institute of Information & Communications Technology Planning & Evaluation (IITP) grant funded by the Korea government (MSIT) (No.2022-0-00953, Self-directed AI Agents with Problem-solving Capability; No.2019-0-00075, Artificial Intelligence Graduate School Program (KAIST)). This material is based upon work supported by the Google Cloud Research Credits program with the award (A696-P323-JC3B-RNWG).

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

## A    TASKS AND DATASETS

In this section, we describe the details of control tasks from D4RL benchmarks (Fu et al., 2020).

**AntMaze**. AntMaze is a navigation task requiring a Mujoco Ant robot to reach a goal location. The datasets are generated by a pre-trained policy designed to reach a goal on different maze layouts. We consider two maze layouts: *medium* and *large*. Datasets are generated by two strategies: *diverse* and *play*. The *diverse* datasets are generated from the pre-trained policy with randomized start and goal locations. The *play* datasets are generated from the pre-trained policy with specific hand-picked goal locations. Task reward is constructed in a sparse setting that gives rewards only when the distance to the goal location is less than a fixed threshold, or all of them give zero.

**Gym-Mujoco locomotion**.  The goal of Gym-Mujoco locomotion tasks is to control the simulated robots (Walker2d, Hopper) such that they can move forward while minimizing the energy cost (action norm) for safe behaviors. We consider two data generation strategies: *medium-expert* and *medium-replay*. The *medium-expert* datasets are generated by mixing equal amounts of expert demonstrations, suboptimal (partially-trained) demonstrations, and *medium-replay* datasets correspond to the replay buffer collected by a partially-trained policy. The task reward is defined as the forward velocity of the torso, control penalty, and survival bonus.

**Robosuite robotic manipulation.** Robosuite robotic manipulation (Zhu et al., 2020) includes different types of tasks with various 7-DoF simulated Hand robots. We use simulated environments with Panda by Franka Emika for the robot in our experiments. We choose the task of lifting a cube object (*lift*) and placing a coke can from table to the target bin (*can*). Datasets are collected by either one proficient teleoperators (*ph*) or 6 teleoperators with varying proficiency (*mh*). Task reward is defined as a sparse reward, and we defer this detail to the original paper.

## B    EXPERIMENTAL DETAILS

**Training details.** Our implementation of Preference Transformer is available at:



**https://github.com/csmile-1006/PreferenceTransformer**



Our model is implemented based on a publicly available re-implementation of GPT in JAX (Frostig et al., 2018)[8]. We use the hyperparameters in Table 2 for all experiments. We use segments of length 100 in all experiments. To improve the stability in training IQL with learned reward function, we normalize the learned reward as follows:

$$\texttt{predicted reward} = \texttt{max timestep} \times \frac{\texttt{learned reward} - (\texttt{max returns})}{(\texttt{max returns}) - (\texttt{min returns})},$$

where `max timestep` is the maximum episode length, and `max timestep` and `min timestep` denote returns of best and worst trajectories in the dataset, respectively. Training time varies depending on the environment, but it takes less than 10 minutes for reward learning in AntMaze environment with 1000 human feedback and takes about an hour to train IQL with learned PT. For training and evaluating our model, we use a single NVIDIA GeForce RTX 2080 Ti GPU and 8 CPU cores (Intel Xeon CPU E5-2630 v4 @ 2.20GHz). We train both reward function and IQL over 8 random seeds.

**Implementation details of PT**. In all experiments, we use causal transformers with one layer and four self-attention heads followed by a bidirectional self-attention layer with a single self-attention head. PT is trained using AdamW optimizer (Loshchilov & Hutter, 2019) with a learning rate of $1 \times 10^{-4}$ including linear warmup steps of 5% of total gradient steps, cosine learning rate decay, weight decay of $1 \times 10^{-4}$, and batch size of 256. For RL training, we use publicly released implementations of IQL[9] and follow the original hyper-parameter settings. For all experiments, we use the same hyperparameters used by the original IQL.

**Evaluation details.** We follow the original settings of IQL in evaluation. We evaluate IQL over 100 rollouts in AntMaze, 10 rollouts in Gym-Mujoco locomotion, and 50 rollouts in Robosuite robotic manipulation. For evaluation metrics, we measure expert-normalized scores introduced in D4RL

---

[8] https://github.com/matthias-wright/flaxmodels
[9] https://github.com/ikostrikov/implicit_q_learning

benchmark: `normalized score` $= 100 \times \frac{\texttt{score} - \texttt{random score}}{\texttt{expert score} - \texttt{random score}}$ for AntMaze and Gym-Mujoco locomotion tasks and success rate for Robosuite robotic manipulation tasks.

**Hyperparameters**. Hyperparameters for PT are shown in Table 2. We remark that PT with more attention layers and more gradient updates can boost the model's performance, but we use the current version with small layers for faster training and evaluation.

Table 2: Hyperparameters of Preference Transformer.

| Hyperparameter | Value |
| --- | --- |
| Number of layers | 1 |
| Number of attention heads | 4 |
| Embedding dimension | 256 |
| (Casual transformer, Preference attention layer) | |
| Batch size | 256 |
| Dropout rate (embedding, attention, residual connection) | 0.1 |
| Learning rate | 0.0001 |
| Optimizer | AdamW (Loshchilov & Hutter, 2019) |
| Optimizer momentum | $\beta_1 = 0.9, \beta_2 = 0.99$ |
| Weight decay | 0.0001 |
| Warmup steps | 500 |
| Total gradient steps | 10K |

**Baselines.** Implementation details of our baselines are as follows:

- **Markovian reward model (MR)**. We use two-layer MLPs with 256 hidden dimensions each. We use ReLUs for the activation function between layers, and we do not use the activation function for the output. Each model is trained by optimizing the cross-entropy loss defined in equation 2 with the learning rate of 0.0003.

- **Non-Markovian reward model (NMR)**. For the non-Markovian reward model, we re-implement CSC Instance Space LSTM (Early et al., 2022) following the architecture specified in the original paper using JAX. We double the hidden dimensions to match the NMR and PT.

**Implementation details of Hopper backflip.** For enabling Hopper backflip, we train the agent in online setup following Lee et al. (2021b). We pre-train the policy $\pi$ using an intrinsic reward for the first 10,000 timesteps, to explore and collect diverse experiences. Then, we train PT using collected behaviors and relabel past experiences in the replay buffer using the learned reward model. This reward learning and relabeling stage is repeated every 10,000 timesteps, and we give 100 queries of human feedback at each stage. We observe that 3 reward learning stages (*i.e.*, 300 queries) are enough to perform double backflips. We use 3 layers and 8 attention heads for PT, and the model is trained for 100 epochs at each stage. The other details are the same as in Table 2.

## C  HUMAN PREFERENCES

**Preference collection.** We collect feedback from actual human subjects (the authors) familiar with robotic tasks. In detail, the human teacher who is given instruction on each task watches a video rendering each segment and chooses which of the two is more helpful in achieving the objective of the agent. Each trajectory segment is 3 seconds long (100 timesteps). If the human teacher cannot decide about the preference over segments, it is allowed to select a neutral option that gives the same preference to each of the two segments.

**Instruction given to human teacher.**

- AntMaze: The first priority is for the ant robot to reach the goal location as soon as possible without wandering or falling. If the ant robot is either left falling, hovering, or moving in the

opposite direction to the goal location, lower your priority to the segment even if the distance to the goal direction is closer than that of the other segment. If the two robots are almost tied on this metric, choose the segment by the distance that the robot has moved.

- Hopper: The hopper robot aims to move to the right as far as possible while minimizing energy costs. If the hopper robot lands unsteadily, lower your priority even if the distance traveled moved during a segment is longer than the other. If the two robots are almost tied on this metric, choose the segment by the distance that the robot has moved.

- Walker2d: The goal of the walker robot is to move to the right as far as possible while minimizing energy costs. If the walker is about to fall or walks abnormally (e.g., walking using only one leg, slipping, etc.), lower your priority to the segment even if the distance traveled moved during a segment is longer than the other. If the two robots are almost tied on this metric, choose the segment by the distance the robot has moved.

- Robosuite: Panda robot arm first grasps the object and carries out actions for a specific purpose (lift a cube or move a coke can to the target bin). Prioritize catching the object between the two. If the two robots are almost tied on this metric, choose by the extent to which the target object has moved.

## D  FULL EXPERIMENTAL RESULTS WITH SCRIPTED TEACHER

Table 3: Averaged normalized scores of IQL on AntMaze, Gym-Mujoco locomotion tasks, and success rate on Robosuite manipulation tasks with different reward functions. We train Preference Transformer (PT) and standard preference modeling with Markovian reward (MR) modeled by MLP or non-Markovian reward (NMR) modeled by LSTM using the same dataset of preferences from scripted teachers and real human teachers. The result shows the average and standard deviation averaged over 8 runs.

| Dataset | IQL with task reward | IQL with preference learning | | | | | |
| --- | --- | --- | --- | --- | --- | --- | --- |
| | | Scripted Teacher | | | Human Teacher | | |
| | | MR | NMR | PT (ours) | MR | NMR | PT (ours) |
| antmaze-medium-play-v2 | 73.88 $\pm$ 4.49 | 64.75 $\pm$ 5.23 | 66.13 $\pm$ 3.36 | 66.13 $\pm$ 3.36 | 31.13 $\pm$ 16.96 | 62.88 $\pm$ 5.99 | 70.13 $\pm$ 3.76 |
| antmaze-medium-diverse-v2 | 68.13 $\pm$ 10.15 | 4.25 $\pm$ 6.27 | 67.00 $\pm$ 5.90 | 68.13 $\pm$ 4.88 | 19.38 $\pm$ 9.24 | 20.13 $\pm$ 17.12 | 65.25 $\pm$ 3.59 |
| antmaze-large-play-v2 | 48.75 $\pm$ 4.35 | 21.00 $\pm$ 14.23 | 10.75 $\pm$ 1.98 | 23.13 $\pm$ 13.10 | 24.25 $\pm$ 14.03 | 14.13 $\pm$ 3.60 | 42.38 $\pm$ 9.98 |
| antmaze-large-diverse-v2 | 44.38 $\pm$ 4.47 | 15.75 $\pm$ 6.32 | 12.00 $\pm$ 3.59 | 44.50 $\pm$ 5.90 | 5.88 $\pm$ 6.94 | 0.00 $\pm$ 0.00 | 19.63 $\pm$ 3.70 |
| antmaze-v2 average | 58.79 | 26.31 | 38.97 | 50.00 | 20.16 | 24.29 | 49.35 |
| hopper-medium-replay-v2 | 83.06 $\pm$ 15.80 | 62.77 $\pm$ 9.36 | 72.33 $\pm$ 0.01 | 94.19 $\pm$ 6.08 | 11.56 $\pm$ 30.27 | 57.88 $\pm$ 40.63 | 84.54 $\pm$ 4.07 |
| hopper-medium-expert-v2 | 73.55 $\pm$ 41.47 | 80.00 $\pm$ 33.06 | 85.97 $\pm$ 37.91 | 39.14 $\pm$ 29.33 | 57.75 $\pm$ 23.70 | 38.63 $\pm$ 35.58 | 68.96 $\pm$ 33.86 |
| walker2d-medium-replay-v2 | 73.11 $\pm$ 8.07 | 65.69 $\pm$ 8.17 | 73.63 $\pm$ 7.35 | 77.08 $\pm$ 9.84 | 72.07 $\pm$ 1.96 | 77.00 $\pm$ 3.03 | 71.27 $\pm$ 10.30 |
| walker2d-medium-expert-v2 | 107.75 $\pm$ 2.02 | 109.95 $\pm$ 0.54 | 110.41 $\pm$ 0.82 | 109.99 $\pm$ 0.63 | 108.32 $\pm$ 3.87 | 110.39 $\pm$ 0.93 | 110.13 $\pm$ 0.21 |
| locomotion-v2 average | 84.37 | 79.60 | 85.56 | 80.10 | 62.43 | 70.98 | 83.72 |
| lift-ph | 96.75 $\pm$ 1.83 | 95.75 $\pm$ 2.71 | 80.00 $\pm$ 19.18 | 92.50 $\pm$ 4.24 | 84.75 $\pm$ 6.23 | 91.50 $\pm$ 5.42 | 91.75 $\pm$ 5.90 |
| lift-mh | 86.75 $\pm$ 2.82 | 92.25 $\pm$ 4.83 | 91.50 $\pm$ 5.42 | 93.00 $\pm$ 3.55 | 91.00 $\pm$ 4.00 | 90.75 $\pm$ 5.75 | 86.75 $\pm$ 5.95 |
| can-ph | 74.50 $\pm$ 6.82 | 62.25 $\pm$ 12.02 | 67.75 $\pm$ 6.80 | 69.75 $\pm$ 9.04 | 68.00 $\pm$ 9.13 | 62.00 $\pm$ 10.90 | 69.75 $\pm$ 5.89 |
| can-mh | 56.25 $\pm$ 8.78 | 59.00 $\pm$ 1.10 | 54.25 $\pm$ 5.57 | 55.25 $\pm$ 6.65 | 47.50 $\pm$ 3.51 | 30.50 $\pm$ 8.73 | 50.50 $\pm$ 6.48 |
| robosuite average | 78.56 | 77.31 | 73.37 | 77.63 | 72.81 | 68.69 | 74.68 |

## E    ADDITIONAL EXAMPLES OF QUERIES

In Figure 8, we provide additional examples of queries to highlight the difference between human and scripted teachers. In the case of the example in Figure 8a, the scripted teacher prefers trajectory 1 since ant is closer to the goal. However, the human teacher prefers trajectory 0 since ant is flipped in trajectory 1. In the case of the example in Figure 8b, the scripted teacher shows a very myopic decision because a hand-designed reward does not capture the context of behavior (*e.g.*, falling down). These examples show the lack of ability to consider multiple objectives, while humans can provide more reasonable feedback by balancing the multiple objectives.

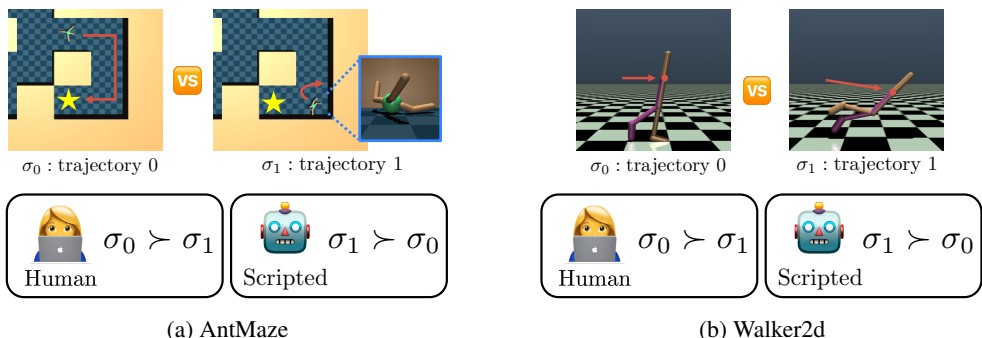

(a) AntMaze

(b) Walker2d

Figure 8:    Illustrations of trajectories from (a) `antmaze-large-play-v2` and (b) `walker2d-medium-replay-v2`. Since hand-designed task reward does not contain all task-relevant information, scripted teacher ignores important events such as flipping and falling down.

## F    DESCRIPTION OF THE VIDEO EXAMPLES

We provide video examples with visualization of the learned importance weight in the supplementary material. For a better visibility, we represent the weight as a color map on the rim of the video, *i.e.*, we highlight frames with higher weights using more bright colors. We observe that the learned importance weights are well-aligned with human intent, as in Figure 3.

## G  EXPERIMENTAL RESULTS USING NON-MARKOVIAN MODELS

We investigate whether our non-Markovian reward function shows better performance with non-Markovian policy and value functions. For each timestep $t$, we augment state $s_t$ by concatenating historical information[10] , and use it as inputs to train policy and value functions. Table 4 shows the results comparing Preference Transformer and NMR. We observe that PT outperforms NMR, which again shows the superiority of our method. However, compared to the results with Markovian policy and value functions in Table 1, the overall performances of all methods are degraded. We expect that this is because the original offline RL algorithm (IQL) assumed Markovian setup and tuned all hyper-parameters under this assumption.

Table 4: Averaged normalized scores of non-MDP IQL with different reward functions on AntMaze and Gym-Mujoco locomotion tasks. Using the same dataset of preferences from real human teachers, we train Preference Transformer (PT) and LSTM-based non-Markovian reward (NMR; Early et al. 2022). The result shows the average and standard deviation averaged over 8 runs.

| Dataset | IQL with preference learning | |
| --- | --- | --- |
| | NMR | PT (ours) |
| antmaze-medium-play-v2 | 54.50 $\pm$ 4.59 | 64.38 $\pm$ 3.16 |
| antmaze-medium-diverse-v2 | 10.50 $\pm$ 13.60 | 53.00 $\pm$ 21.56 |
| antmaze-large-play-v2 | 14.83 $\pm$ 5.78 | 25.63 $\pm$ 4.78 |
| antmaze-large-diverse-v2 | 0.00 $\pm$ 0.00 | 8.13 $\pm$ 3.27 |
| antmaze-v2 total | 19.96 | 37.79 |
| hopper-medium-replay-v2 | 15.80 $\pm$ 26.11 | 55.77 $\pm$ 31.09 |
| hopper-medium-expert-v2 | 55.80 $\pm$ 34.34 | 79.84 $\pm$ 29.32 |
| walker2d-medium-replay-v2 | 41.91 $\pm$ 32.58 | 67.06 $\pm$ 13.39 |
| walker2d-medium-expert-v2 | 94.20 $\pm$ 29.41 | 104.97 $\pm$ 10.63 |
| locomotion-v2 total | 51.92 | 76.91 |

---

[10]Specifically, we provide a predicted reward of the previous timestep $\hat{r}_{t-1}$ as an additional input.

