# OpenReview forum: "Preference Transformer: Modeling Human Preferences using Transformers for RL"
_ICLR.cc/2023/Conference — ICLR 2023 poster_

### Official Review · Reviewer_v17x · 2022-10-14

**Confidence:** 3
**Correctness:** 3
**Technical Novelty And Significance:** 3
**Empirical Novelty And Significance:** 3
**Recommendation:** 5

**Clarity, Quality, Novelty And Reproducibility:**

As mentioned, the authors need to clarify some issues and make some assumptions explicit:
- $r_t$ is only defined inline in the text, which makes it hard to determine how many states are available for the reward approximation. To my understanding, 1 to H states are available, depending on the position of the state, action pair in the sequence?
- Compared trajectory segments always have same length (H), but trajectories can be longer than H? This would be a deviation from most other PbRL algorithms and poses some more questions. Namely, usually PbRL usually assumes that compared trajectories are starting in the same state. Otherwise, comparability may not be given, because the frame of reference differs (two different "problems").
- Due to the same length, weighted sum and weighted average are equivalent (up to a multiplicative factor)

These clarifications are important, because if my understanding is correct, and important issue may arise: For several state, action rewards, no history information is available. This means, the algorithm can only learn a conventional, Markovian reward. In case, very high numbers of preferences are used, this will affect many states, potentially leading to a Markovian solution in general. Therefore, it is a bit in question if the reason for the observed improvements are really due to the non-Markovian modelling. To resolve this issue, all rewards should have access to the same number history states. Formally, this would allow an N-state history MDP formalization.

The presentation is good, only colors in Fig.4 should not be reused (scripted, human). The algorithm is novel and interesting, especially the combination with a transformer-based model. Reproducibility should be possible, given source code an hyperparameters are given.


**Strength And Weaknesses:**

The consideration of non-Markovian trajectory preferences is an important issue in PbRL, because the Markov property is usually not assumed to hold in this setting. Therefore, non-Markovian PbRL is important to move from synthethic evaluations to real world applications. The paper is mostly well written, but some assumptions are hidden and some clarifications are required. (See next Section) The evaluation covers a sufficiently divers set of domains and real, human evaluations are usually to be preferred over pure synthethic trials. However, the evaluation is lacking in several parts:

- Evaluation is only done in terms of reward. However, user preference may not be aligned with the given reward function. In fact, if it is, the domain does not require a non-Markovian approach. Therefore, evaluation should be performed in terms of preference agreement of the resulting policy. Furthermore, the number of queries is usually also deemed an important metric, because human evaluations are costly.
- The scripted evaluation uses a Markovian reward, but the NMR and PT approaches are still able to outperform the MR variant in several domains. This needs to be better explain and evaluated. In fact, a non Markovian reward should be used for the evaluation.
- The RL algorithm used to maximize the reward assumes a common MDP structure. This means, it will likely not correctly consider the non-Markovian property or the weighting factors. Considering it works, it may be that the learned reward "collapses" to an Markovian variant (also see next Section concerning this issue). To resolve this issue, it should be validated that the learned reward cannot be approximated with a Markovian definition and/or a PO-MDP or N-state MDP RL algorithm should be used.



**Summary Of The Paper:**

The authors introduce a novel method for Preference-based Reinforcement Learning, which targets the problem of non-Markovian rewards. Namely, that human trajectory preferences cannot be assumed to be based on a sum of Markovian rewards, as expected in common RL and PbRL settings. The method learns non-Markovian rewards, based on the assumptions, that trajectory segment returns can be defined as weighted sum over history dependent state, action rewards. The learning algorithm uses a transformer architecture to these ends. Besides the novel PbRL algorithm, the authors contribute an empirical evaluation, demonstrating the advantages on 5 different domains (with 2-4 variants each).

**Summary Of The Review:**

The paper is a nice read and the method is novel and interesting. However, the stated issues with a main property of the algorithm prevent an accept. Although, to be fair, the authors claims are not directly related to the property and only refer to "real human preferences" and are therefore mostly substantiated. Some issues concerning the evaluation metrics and clarity remain, but can likely be resolved without substantial time effort.

---

> ### Author Response · Authors · 2022-11-16
> **Response to Reviewer v17x**
>
> Dear Reviewer v17x,
>
> We sincerely appreciate your valuable comments. We found them extremely helpful in improving our draft. We have updated our revision based on your comments and colored it in red. We address each comment in detail, one by one below.
>
>
> **Q1. Human evaluation**
>
> **A1.** Thank you for the valuable suggestions. Following your suggestion, we have conducted an additional evaluation based on human judgments. Specifically, we generate two trajectories from agents from two different rewards, and then a human evaluator determines which agent better completed the task. We remark that trajectories are anonymized for fair comparisons. In our experiments, we find that human evaluators prefer agents from PT compared to agents from MR and NMR, e.g., PT achieves 71.2% preference rate while NMR achieves 16.3%. These results indicate that PT is more aligned with human preferences. We have included new results in Section 5.2 of the revised draft.
>
>
> **Q2. Why do PT and NMR outperform MR in scripted teacher settings?**
>
> **A2.** We expect this is because the non-Markovian formulation (i.e., PT and NMR) can induce a better-shaped reward by utilizing historical information (i.e., sequence of previous states and actions). Also, the non-Markovian formulation can be interpreted as a generalized version of Markovian formulation since it can learn Markovian rewards by ignoring the past history in the inputs. We have clarified this in the revised draft.
>
>
> **Q3. Non-MDP RL algorithm should be used for evaluation.**
>
> **A3.** As you pointed out, non-MDP RL algorithms would be more suitable to evaluate the non-Markovian reward. However, we utilized an MDP RL algorithm since the original offline RL algorithm (IQL) assumed MDP setup and tuned all hyper-parameters under this assumption. Since offline RL algorithms are usually sensitive to hyper-parameters, we found that training non-Markovian models (e.g., providing history information to policy and value functions) using IQL is very unstable. To support this, we have included offline RL experiments with the non-MDP policy and value functions (i.e., using history information of the last N states as you suggested) in Appendix G of the revised draft. As shown in Table 4, we observed that PT still outperforms all baselines with non-Markovian policy and value functions but the overall performances of all methods are degraded. We expect that more hyper-parameter tuning is required and thus will update the results with the further tuning of the hyperparameters. We have highlighted this in the revised draft.
>
> **Q4. Editorial comments / clarifications**
>
> **A4.**
> 1. Reward inputs: Your understanding is correct. Recent $H$ state-action pairs are provided as input to the reward function
> 2. Weighted sum vs. weighted average: Yes, weighted sum and weighted average are equivalent.
> 3. Figure 4: Thank you for the detailed comments. We have updated Figure 4 in the revised draft for better visualization.
>
> We have clarified the above in the revised draft.
>
> **Q5. Does non-Markovian reward “collapse” to a Markovian?**
>
> **A5.** Except for initial states ($t=1$), historical information (i.e., sequences of visited states and executed actions) is always available, and thus we don’t think that our method converges to Markovian reward.
>
> **Q6. Trajectory segment.**
>
> **A6.** Several recent work on preference-based RL [2, 3, 4, 5] (including our work) usually compares trajectory segments from different states (but with same length) and it has been observed that humans can provide useful signals (i.e. preferences between two segments). This is because trajectory segments from different states can be interpreted as different behaviors, not totally different problems.
>
> **References**
>
> [1] Scott Fujimoto and Shixiang Shane Gu, A Minimalist Approach to Offline Reinforcement Learning, NeurIPS 2021.
>
> [2] Paul F Christiano et al., Deep Reinforcement Learning from Human Preferences, NIPS 2017.
>
> [3] Kimin Lee et al., PEBBLE: Feedback-Efficient Interactive Reinforcement Learning via Relabeling Experience and Unsupervised Pre-training, ICML 2021.
>
> [4] Jongjin Park et al., SURF: Semi-supervised Reward Learning with Data Augmentation for Feedback-efficient Preference-based Reinforcement Learning, ICLR 2022.
>
> [5] Donald Joseph Hejna III and Dorsa Sadigh., Few-Shot Preference Learning for Human-in-the-Loop RL, CoRL 2022.

---

> > ### Comment · Reviewer_v17x · 2022-11-22
> > **Response to Authors**
> >
> > Thanks for the detailed answers. I can follow your argumentation and the new results are a good extension. However, a more systematic (re-)evaluation is required to substantially raise the score. Although, i will not argue against acceptance and considering my evaluation was only borderline-reject before, i would raise the score by 1 point in case may review is the only reason for rejection.

---

> > > ### Author Response · Authors · 2022-11-22
> > > **Thank you for providing additional feedback**
> > >
> > > Thank you for the positive response. We are happy to hear that our response and additional results address your concerns.
> > >
> > > Could you elaborate more about `a more systematic (re-)evaluation`? We wonder if this meant the human A/B testing we did in the revised version or a different evaluation. We will do our best to include those evaluation results in the final version of our manuscript.
> > >
> > > Best,
> > >
> > > Authors

---

> > > > ### Comment · Reviewer_v17x · 2022-11-23
> > > > **Systematic (re-)evaluation**
> > > >
> > > > I was simply referring to the more exhaustive evaluations already discussed (non markovian rewards, PO-MDP algorithm, reward collapse) Your argumentation is sound, but still an "educated guess", no formal or empirical evidence. Although, i understand this is out of scope for this submission due to the required, additional experiments and hyperparameter tuning.

---

> > > > > ### Author Response · Authors · 2022-11-23
> > > > > **Thank you for the response**
> > > > >
> > > > > Thank you again for answering our question. We believe your valuable suggestions and comments would strengthen our paper. We will try to incorporate all your comments in the final version as much as possible. If you have any remaining suggestions or concerns, please let us know.

---

### Official Review · Reviewer_EobQ · 2022-10-25

**Confidence:** 3
**Correctness:** 3
**Technical Novelty And Significance:** 3
**Empirical Novelty And Significance:** 3
**Recommendation:** 6

**Clarity, Quality, Novelty And Reproducibility:**

In Table 1 — is there an explanation for why NMR performs so poorly in hopper-medium-replay-v2?

You show the agreement between humans and scripted feedback, but I am wondering about the inter-annotator agreement amongst the authors.  Also, how does the agreement change as the model improves?

How was the feedback signal formed from the human annotators?  Was it majority vote?  What about ties?


**Strength And Weaknesses:**

+The paper is well written

+The transformer approach naturally lends itself to credit assignment, which is important for human preferences as human are significantly influenced by surprising events.

-Given an advantage of this approach is feedback-efficient, it would have been nice to have compared against PEBBLE given that is designed for efficiency.


**Summary Of The Paper:**

This paper describes an approach for learning a reward function in preference-based reinforcement learning based on transformers.  This provides the advantages that the model allows credit assignment within the behavior trajectory to correctly weight significant state/actions, and that less feedback samples that has typically been required to learn the reward.

**Summary Of The Review:**

Reward learning for preference-based RL is receiving increasing interest, so this is timely.  Transformers are a natural way to accommodate the problem of credit assignment over a trajectory.

---

> ### Author Response · Authors · 2022-11-16
> **Response to Reviewer EobQ**
>
> Dear Reviewer EobQ,
>
> We sincerely appreciate your valuable comments. We found them extremely helpful in improving our draft. We have updated our revision based on your comments and colored it in red. We address each comment in detail, one by one below.
>
> **Q1. Comparison with PEBBLE [1].**
>
> **A1.** Thank you for the interesting suggestion. We think that Preference Transformer (PT) can be combined with PEBBLE and further improve the feedback-efficiency since PT and PEBBLE have orthogonal contributions (i.e., we focus on reward learning while PEBBLE focuses on RL training). However, implementing an online setup requires more time and resources, and thus we would like to leave it as a future investigation.
>
>
> **Q2. Why NMR performs poorly in hopper-medium-replay-v2?**
>
> **A2.** Thank you for pointing this out. We found that NMR is sensitive to hyperparameters and random seeds in Gym-Mujoco hopper tasks, possibly due to the early termination. We have further tuned hyperparameters (e.g., hidden dimension of the LSTM) of NMR and updated the results in Table 1 of our revised draft. In the updated results, NMR achieves better performance than MR in hopper-medium-replay-v2, but PT still outperforms NMR significantly in most tasks.
>
>
> **Q3. Handling multiple annotators.**
>
> **A3.** We would like to clarify that each task is annotated by a single human expert to avoid issues coming from multiple annotators (e.g., how to combine the feedback). We think investigating a setup with multiple annotators is an interesting direction to explore in future studies.
>
> **References**
>
> [1] Kimin Lee et al., PEBBLE: Feedback-Efficient Interactive Reinforcement Learning via Relabeling Experience and Unsupervised Pre-training, ICML 2021.

---

### Official Review · Reviewer_gNsz · 2022-10-25

**Confidence:** 3
**Clarity, Quality, Novelty And Reproducibility:** See above.
**Correctness:** 3
**Technical Novelty And Significance:** 2
**Empirical Novelty And Significance:** 2
**Recommendation:** 6

**Strength And Weaknesses:**

**Strengths.**

Except for details, overall the paper is well written.

The preference RL setting is interesting,
and,
if in deed the rewards are non-Markovian,
the idea of Transformers is well motivated.

So there is a conceptual as well as empirical contribution.


**Weaknesses and improvement points:**

I think the precise contribution over previous work needs to be clarified:
* From a first look at the paper, I had these questions: does a preference transformer already exist, but just for Markovian, unweighted rewards, and the contribution is just to extend it to the non-Markov, weighted setting? Or is the contribution, to introduce the preference transformer thing altogether, and directly in the general from (non-M. ...)?
* When looking into it in more detail, I understand it as follows: it seems (Early 2022) is closest, since they also address preference RL with non-Markov rewards, and with a similar relation between rewards and preferences, but the difference is that (Early 2022) (1) use LSTM instead of Transformers and (2) don't allow for weighing. This, in particular, should be made more clear.

Regarding experiments, there need to be some clarifications, and some results are counterintuitive:
* For the tasks that are studied, are the true reward functions actually Markovian or not? I don't find that info. Or is there no true reward for some tasks?
* Sec 5.4: if the reward is actually Markovian, then the much simpler MR baseline model (which assumes a Markovian reward via an MLP) actually has *no approximation error in that sense (i.e., up to functional approximation, the dependencies between the variables can in principle perfectly be captured by this model class)*, but is much simpler in terms of capacity (so, should be more data efficient). I'm really surprised that nonetheless PT outperforms it. Can you explain why PT outperforms it?

**Minor points:**

* Note that non-Markovian rewards can be turned into markovian my changing the state.
* How does it relate to V/Q value function? It's not necessary, but 1-2 sentences could be interesting.

**Summary Of The Paper:**

Given: just preferences over agent trajectories (and not rewards),
and the "underlying" reward being non-Markovian and weighted.

Approach: the authors propose the Preference Transformer model which learns the "latent reward" based on the whole past being fed into a Transformer to output current reward and weight. Then from this reward they can classically train an agent.

The contribution includes this architecture as well as experiments


**Summary Of The Review:**

Overall, it is an interesting idea to use Transformers for non-Markovian reward and weighting, as part of preference RL. The writing overall is OK-good, though some clarifications are needed.

The main weaknesses for me are that the contribution is interesting but rather limited compared to (Early 2022) (replace LSTM by Transformer and add weighting). Additionally, I'm not sure how often actually non-Markovian rewards are needed, since they can be mitigated by good state representations. And, in the experiment, if the reward is Markovian, I didn't understand why nonetheless the non-Markovian method performs better.

When ignoring significance/size of contribution, I lean accept; when not ignoring it, I lean reject. In favor of the doubt, overall I lean accept.

---

> ### Author Response · Authors · 2022-11-16
> **Response to Reviewer gNsz**
>
> Dear Reviewer gNSz,
>
> We sincerely appreciate your valuable comments. We found them extremely helpful in improving our draft. We have updated our revision based on your comments and colored it in red. We address each comment in detail, one by one below.
>
> **Q1.  Main contribution of our work.**
>
> **A1.** Our main contribution is on both (a) proposing a new preference model based on the weighted sum of non-Markovian rewards and (b) introducing Preference Transformer for modeling the proposed preference model. We have emphasized this in the revised draft. Thank you for the good question.
>
>
> **Q2. Difference between our work and Early et al., 2022 [1].**
>
> **A2.** We first remark that Early et al., 2022 [1] did not consider preference-based learning (i.e., learning a reward from preferences). They focused on a regression setup where a scalar value is given as human feedback. They assumed that scalar-valued feedback corresponds to partial returns (sum of non-Markovian rewards) and proposed LSTM-based architecture to infer the non-Markovian reward from scalar-valued feedback. In short, problem setup is quite different since we focus on learning from preferences (comparison between behaviors). Also, as you pointed out, we use transformers instead of LSTM and consider a weighted sum. Finally, their approach was only evaluated with synthetic feedback, while our method is tested with both synthetic and real human feedback.
>
>
> **Q3. Task reward used for evaluation.**
>
> **A3.** For all tasks, we utilized the task rewards from the original benchmark and described details in Appendix A of the submitted draft. Even though they are defined as Markovian rewards, we used them since (1) task rewards are defined to measure the human-interpretable quality (such as distance to the goal and velocity of agents) and (2) efficient evaluation becomes possible.
> To measure the alignment with real human preferences, we have conducted human evaluations in the revised draft. Specifically, we generate two trajectories from agents trained with two different rewards, and then a human evaluator determines which agent better completed the task. We remark that trajectories are anonymized for fair comparisons. In our experiments, we find that human evaluators prefer agents from PT compared to agents from MR and NMR, showing PT is more aligned with human preferences. We have included new results in Section 5.2 of the revised draft.
>
> **Q4. Why is PT better than MR in a scripted teacher setting?**
>
> **A4.** We expect this is because PT can induce a better-shaped reward by utilizing historical information (i.e., sequence of previous states and actions). Also, the non-Markovian formulation (PT) can be interpreted as a generalized version of Markovian formulation (MR) since it can learn Markovian rewards by ignoring the past history in the inputs. We have clarified this in the revised draft.
>
>
> **Q5. Discussions on V / Q value functions**
>
> **A5.** Thank you for the good questions.
> With the non-Markovian reward, the V / Q value functions also depend on the past history, i.e., they would be non-Markovian. Due to this reason, it would be more suitable to use non-Markovian value functions to evaluate the non-Markovian reward. However, we utilized an MDP RL algorithm since the original offline RL algorithm (IQL) assumed MDP setup and tuned all hyper-parameters under this assumption. Since offline RL algorithms are usually sensitive to hyper-parameters, we found that training non-Markovian models (e.g., providing history information to policy and value functions) using IQL is very unstable. For supporting results, we have included offline RL experiments with the non-Markovian policy and value functions (i.e., using historical information of the last N states) in Appendix G of the revised draft. As shown in Table 4, we observed that PT still outperforms all baselines but the overall performances of all methods are degraded when we use the non-Markovian policy and value functions. We expect that more hyperparameter tuning is required and thus will update the results with further tuning. We have highlighted this in the revised draft.
>
>
> **References**
>
> [1] Joseph Early et al., Non-Markovian Reward Modelling from Trajectory Labels via Interpretable Multiple Instance Learning, NeurIPS 2022.
>
> [2] Paul F Christiano et al., Deep Reinforcement Learning from Human Preferences. NIPS 2017.
>
> [3] Kimin Lee et al., PEBBLE: Feedback-Efficient Interactive Reinforcement Learning via Relabeling Experience and Unsupervised Pre-training, ICML 2022.

---

### Official Review · Reviewer_WbtG · 2022-10-25

**Confidence:** 2
**Correctness:** 3
**Technical Novelty And Significance:** 3
**Empirical Novelty And Significance:** 3
**Recommendation:** 8

**Clarity, Quality, Novelty And Reproducibility:**

The writing is clear and easy to follow.
The proposed preference predictor based on non-Markovian assumption is novel and effective.
The released code seems easy to run and install.


**Strength And Weaknesses:**

Strengths:
1. This paper is well-written and easy to understand.
2. The proposed new preference predictor that depends exponentially on the weighted sum of non-Markovian rewards is reasonable and effective.
3. Conducted Experiments are well-designed. Preference Transformer are evaluated both on real human preferences and synthetic preferences. The authors demonstrate the relationship between them and show that existing benchmarks based on synthetic preferences may not be enough.
Weaknesses:
1. It would be interesting to see justifications for the architecture selection. Did you compare the GPT with other transformer architectures in experiments?
2. In Section 5.2, the authors demonstrate that capturing long-term information is the reason why PT is more competitive than other baseline models, but LSTM-based NMR could also capture temporal dependencies. There should be detailed explanations to clarify the demonstration.


**Summary Of The Paper:**

This paper presents a transformer-based neural architecture, Preference Transformer, to model human preferences in preference-based reinforcement learning. Specifically, this work adopts the assumption of non-Markovian rewards and utilizes transformer-based architecture in modeling sequential data to capture temporal dependencies in human decisions and infer critical events in a trajectory.

**Summary Of The Review:**

This paper introduces a novel transformer-based architecture to model human preferences under the non-Markovian assumption in preference-based reinforcement learning. The overall framework seems to be effective and performs well according to the experiment results. Though some of the claims have minor issues, the overall paper is well-written and easy to follow.

---

> ### Author Response · Authors · 2022-11-16
> **Response to Reviewer WbtG**
>
> Dear Reviewer WbtG,
>
> We sincerely appreciate your valuable comments. We found them extremely helpful in improving our draft. We address each comment in detail, one by one below.
>
> **Q1. Justification for the architecture selection.**
>
> **A1.** We utilized the GPT causal transformer model since it has been observed that this architecture is very effective in RL domains [1, 2]. We agree that utilizing other transformer architecture is interesting, but we would like to leave it as a future investigation.
>
> **Q2. Why PT outperforms LSTM-based NMR?**
>
> **A2.** As you pointed out, LSTM-based NMR could also capture temporal dependencies. However, we found that transformer-based architecture is more effective in capturing temporal dependencies than LSTM-based architecture. We expect that this is because transformers do not compress the histories into hidden embeddings, unlike LSTM. We also remark that this finding is consistent with the recent success of transformers over LSTM (or RNN) in sequential modeling.
>
> **References**
>
> [1] Lili Chen et al., Decision Transformer: Reinforcement Learning via Sequence Modeling, NeurIPS 2021.
>
> [2] Michael Janner et al., Offline Reinforcement Learning as One Big Sequence Modeling Problem, NeurIPS 2021.

---

### Public Comment · ~Mudit_Verma2 · 2022-11-09
**Comment on the extension to Non Markovian Rewards**

The paper extends the modeling of human binary preferences with Non Markovian rewards, and then use Transformers to capture the weights while computing rewards for a history. Similar work has been done in the past : https://arxiv.org/abs/2210.09151 where such weights are used as a soft prior to best approximate the possible markovian reward.

---

> ### Author Response · Authors · 2022-11-16
> **Response to Mudit Verma**
>
> Thank you for sharing your work [1]. We couldn’t find this work because it was uploaded on arXiv after the ICLR submission deadline. Your idea of utilizing transformer architectures for preference-based reward learning is interesting, but we think there are several differences between our work and your work. For example, your work assumes a Markovian reward model and leverages (non-Markovian) transformer networks for reward learning while we directly design the reward model to be non-Markovian. In addition, your method regularizes the reward to be aligned with the self-attention weights of the transformer, while our method utilizes those weights for modeling a preference model. Finally, your method is tested on a simple grid world while our method is tested on more complex domains.
>
> **References**
>
> [1] Mudit Verma and Katherine Metcalf, Symbol Guided Hindsight Priors for Reward Learning from Human Preferences, arXiv preprint arXiv:2210.09151, 2022.

---

### Author Response · Authors · 2022-11-16
**General Response**

Dear reviewers and area chair,

We sincerely appreciate your time and effort in reviewing our manuscript.
Our work proposes a generalized framework for preference-based RL and a novel neural architecture for modeling this. The reviewers highlighted the novelty of our method (WbtG, v17x), empirical performance (WbtG, gNsz), and clear presentation (WbtG, gNsz, EobQ, v17x).

In response to the questions and concerns you raised, we have carefully revised and enhanced the manuscript with the following additional experiments and discussions:
- Highlighting our contributions (Section 1)
- Improving the method description (Section 4.2)
- Additional experimental results with human evaluation (Section 5.2)
- Detailed explanation for the results in scripted settings (Section 5.4)
- Additional experimental results using non-Markovian models (Appendix G)

These updates are temporarily highlighted in "red" for your convenience to check.

With valuable feedback from the reviewers, we believe that our work has become stronger and could deliver the benefits of our generalized preference modeling and the proposed Preference Transformer architecture in preference-based RL to the audience of ICLR.

Thank you very much,

Authors.

---

### Decision · Program_Chairs · 2023-01-20

**Decision:**

Accept: poster

**Justification For Why Not Higher Score:**

The idea presented in this paper is novel and interesting, however, some more clarification and explanations are needed to make the paper in a better shape, so the AC do not think it can be leveled up to a Spotlight paper.

**Justification For Why Not Lower Score:**

All the reviewers are positive about the paper. The method presented is novel, and one reviewer also commented that the contribution is timely, the reviewer who gave a score of 5 also agrees to converging to positive outcome, so the AC decides to recommend acceptance.

**Metareview: Summary, Strengths And Weaknesses:**

This paper presents Preference Transformer that uses a transformer architecture to model human preferences in preference-based reinforcement learning.

After author rebuttal, it received scores of 5668. The reviewer who gave a score of 5 also clarified that he/she will not argue for rejection. All the reviewers agree that the proposed model for prefrence-based RL is novel, the empirical performance is good, and the presentation is clear. On the other hand, some more clarification and explanations on experimental results are needed, e.g., why PT and NMR outperform MR in scripted teacher settings.

Overall, the AC thinks that the proposed method in this paper is novel and interesting, therefore, recommends acceptance of the paper.

**Note From Pc:**

if the above contains the word "oral" or "spotlight" please see: "oral" presentation means -> notable-top-5% and "spotlight" means -> notable-top-25%. As stated in our emails, we are disassociating presentation type from AC recommendations